# Sustainable Environmental Assessment of Waste-to-Energy Practices: Co-Pyrolysis of Food Waste and Discarded Meal Boxes

**DOI:** 10.3390/foods11233840

**Published:** 2022-11-28

**Authors:** Gang Li, Tenglun Yang, Wenbo Xiao, Jiahui Wu, Fuzhuo Xu, Lianliang Li, Fei Gao, Zhigang Huang

**Affiliations:** 1School of Artificial Intelligence, Beijing Technology, and Business University, No.11 Fuchenglu, Haidian District, Beijing 100048, China; 2Beijing Key Laboratory of Quality Evaluation Technology for Hygiene and Safety of Plastics, College of Chemistry and Materials Engineering, Beijing Technology and Business University, No.11 Fuchenglu, Haidian District, Beijing 100048, China; 3School of Food and Health, Beijing Technology and Business University, No.11 Fuchenglu, Haidian District, Beijing 100048, China

**Keywords:** food waste, co-pyrolysis, synergistic, activation energy, recycling

## Abstract

The reuse of biomass waste is conducive to the recovery of resources and can solve the pollution problem caused by incineration and landfill. For this reason, the thermogravimetric analyzer (TGA) was used to study the pyrolysis of the mushroom sticks (MS) and discarded meal boxes at different heating rates (10 °C·min^−1^, 20 °C·min^−1^, 30 °C·min^−1^). The statistical analysis showed that the factors of pyrolysis temperature and particle size had a greater effect, while the heating rate was significant. The TGA revealed that the maximum weight loss rate of the co-pyrolysis of MS and discarded meal boxes increased with the rise of the heating rate, the temperature at which the pyrolysis started and ended increased, and the thermal weight loss displayed a hysteresis phenomenon. By comparing the theoretical heat weight loss curves with the experimental curves, a synergistic effect of the co-pyrolysis process between MS and discarded meal boxes was demonstrated, and the co-pyrolysis process resulted in a reduction in the solid residue content of the products. The Coats-Redfern method was used to fit the pyrolysis process of MS and discarded meal boxes, which applied the first-order kinetic model to describe the main process of pyrolysis and obtained the reaction activation energy between 43 and 45 kJ·mol^−1^. The results indicated that co-pyrolysis of MS and discarded meal boxes could decrease the activation energy of the reaction, make the reaction easier, promote the degree of pyrolysis reaction, reduce the generation of pollutants, and provide a theoretical basis for the recycling and energy utilization of MS and discarded meal boxes.

## 1. Introduction

The strong growth of food consumption and the rapid development of the food industry have led to the accumulation of large amounts of food waste, which caused serious environmental, social, and economic issues [1,2]. There is a progressive relationship between food, food waste, and sustainability. A large amount of food needed every day leads to large numbers of food waste, and uncontrolled disposal causes serious environmental issues. Therefore, the effective disposal of food waste will realize a sustainable positive impact on the environment. Food waste is a valuable source of organic carbon, which can be converted into high-value-added products and a variety of industrial-grade chemicals through biological processes to achieve the recycling of food waste [3,4]. The concept of food sustainability is included in the concept of environmental sustainability. The sustainability of the process of recycling food waste from the agri-food production chain and the use of biotechnology to ensure the recovery of bioactive compounds from food waste are just a few aspects of the concept of food sustainability [5]. As a result, turning food waste into energy or products not only increases revenue but also addresses the issue of waste disposal, promoting environmental sustainability [6]. Realizing the recycling of food waste and transforming the linear economy into a circular economy has broad development prospects and application value [7]. In addition, the value of food waste (such as fruit and vegetable waste) provides an opportunity to use the extraction of food waste and to develop valuable functional foods [8]. Food waste is converted into fertilizer through composting. The composting process produces CO_2_, NH_3_, and H_2_O as by-products, which lead to the recovery of the mineral nutrients N, P, and K [9]. As a largely agricultural country, the annual output of bamboo, straw, reed, and wheatgrass in China exceeds one billion tons, which are inexhaustible energy sources [10]. Meanwhile, China is the world’s largest country in edible fungus cultivation, the total output of edible fungi was 40.6143 million tons in 2020, and the output scale exceeded 40 million tons for the first time [11,12]. Mushroom sticks (MS) are the cultivation substrate of edible fungi, which are rich in protein, polysaccharides, and minerals [13,14]. At present, MS is widely used in animal feed, soil improvement, secondary cultivation of edible fungi, energy materials, etc. Due to the relatively low content of carbon and hydrogen and the high content of oxygen, the pyrolysis of MS will produce a large amount of carbon deposition. At the same time, the by-products produced are underutilized. In addition, most MS were directly burned, stacked, or buried, which have not been effectively utilized, causing serious environmental pollution [15,16,17].

Due to the impact of the COVID-19 pandemic, the food delivery industry and online shopping in China are quickly developed, which inevitably led to a large number of discarded meal boxes made of polypropylene (PP), which is difficult to degrade, recycle, and utilized forming “white pollution” [18,19,20,21]. According to statistics, global plastic production in 1950 was 2 million tons [22], but it increased to nearly 400 million tons in 2020 [23], with an increase in nearly 200 times. In 2021, the production recovery volume of plastic waste in China was nearly 19 million tons, accounting for only 31% of the production volume [24]. There are currently several different ways to discard meal boxes, including landfilling, incineration, recycling, and composting [25,26]. Nevertheless, the traditional methods need a lengthy decomposition process, large investment, high cost and are difficult to recycle, which leads to serious environmental deterioration [27].

According to the characteristics of the high oxygen content of MS and high hydrogen content of PP, co-pyrolysis is chosen as an effective method with great potential [28,29]. Co-pyrolysis is compared with other pyrolysis processes, such as catalytic pyrolysis and microwave pyrolysis. Catalytic pyrolysis requires the addition of catalysts to produce and enhance pyrolysis products, resulting in inefficiency and waste [30]. Microwave pyrolysis has high costs and difficult heating rate control [31]. However, co-pyrolysis as a promising method is simple, efficient, does not require any solvents or catalysts, and can react without additional hydrogen [28]. Compared with current conventional treatments, co-pyrolysis can degrade plastics and biomass at a lower cost and effectively improve product quality and pollutant control [32,33]. Meanwhile, pyrolysis products can be further processed and applied to all aspects of life [34,35]. It has been proved that the co-pyrolysis of biomass and plastics can enhance the quality of bio-crude [36,37]. Plastic co-pyrolyzed with MS can balance the oxygen content in the product to improve the yield and quality of the bio-oil and the pyrolysis efficiency [38,39,40]. In addition, the energy and environmental problems caused by civilization and industry are addressed through co-pyrolysis. The procedures employ biomass waste, such as animal and agricultural waste, as well as hazardous waste materials, such as waste tires, plastic, and medical waste, as feedstock to create gas, char, and pyrolysis oil for energy generation. Utilizing hazardous materials as co-pyrolysis feedstock lowers the amount of toxic waste dumped into the environment, decreases the likelihood of soil and water contamination, as well as replaces fossil fuels, a non-renewable feedstock [41]. Hassan et al. (2016) found that the yield of the liquid bio-oil prepared by co-pyrolysis can be increased by up to 22% compared to rapid pyrolysis under the same conditions [28]. Ozsin et al. (2018) analyzed the kinetics of co-pyrolysis of cherry seeds and polyvinyl chloride (PVC) and found that the activation energy of co-pyrolysis was significantly reduced compared with that of cherry seeds alone. By comparing the theoretical and experimental TG values, there is a synergy between cherry seeds and PVC [42].

Thus, a hypothesis was proposed that there might be a synergistic effect between MS and PP co-pyrolysis. Due to the synergistic effect of co-pyrolysis of biomass and plastic, as well as the few studies on co-pyrolysis of MS and PP at different heating rates, the following research was carried out in this paper: (1) elemental analysis and thermogravimetric experiment were conducted on MS and discarded meal boxes to determine the weight loss data of individual pyrolysis. (2) Analyze and compare the weight loss data of the separate pyrolysis of MS and discarded meal boxes and study the influence of heating rate on the separate pyrolysis of MS and discarded meal boxes. (3) Co-pyrolysis experiments were conducted on mixed samples of MS and discarded meal boxes to analyze the pyrolysis performance parameters of MS and waste meal boxes under different heating rates, and the theoretical value of thermal weight loss was compared with the experimental value to further study whether the co-pyrolysis of MS and discarded meal boxes had a synergistic effect. (4) The kinetic model was established, and the linear fitting of the experimental curve was performed to calculate the activation energy and frequency factor of the co-pyrolysis of MS and discarded meal boxes.

## 2. Materials and Methods

### 2.1. Samples Preparation

MS was obtained from Pingquan County Huixin Mushroom Industry Co., LTD. (Chengde, China). The discarded meal box made of polypropylene (PP) was purchased from Xingtai Plastic Material Co., Ltd. (Xingtai, China). All samples were ground to small particles less than 2 mm. The raw materials were dried at 105 °C for 24 h to a consistent weight, after which they were filtered through a 60-mesh sieve, bagged, and stored in a dryer.

### 2.2. Test Device and Operating Parameters

In order to explore the effect of the warming rate on the separate pyrolysis and co-pyrolysis of MS and discarded meal box, a Thermogravimetric analyzer (TGA5500, Shanghai Simaio Analytical Instrument Co., Ltd., Shanghai, China) was used to analyze thermogravimetric characteristics at the three different heating rates of 10, 20, 30 °C·min^−1^. MS and PP were blended at a weight ratio of 1:1 to prepare the mixed sample named SP11. During the experiment, 7±1 mg of the treated sample using analytical scales was placed into the platinum crucible. The high-purity helium gas (99.999%) (25 mL·min^−1^) was used as protective gas, and the experimental temperatures ranged from room temperature to 800 °C. The thermogravimetric analyzer synchronously records the thermal weight curve (TG) and the differential thermal weight curve (DTG) during the reaction. Each sample was tested three times within the experimental error of less than ±3% in weight loss measurement.

### 2.3. Analysis of the Thermodynamics Model

First-order dynamic equations are applied to biomass and plastic pyrolysis [43]. The pyrolysis dynamics equation is shown in Equation (1):(1)$$\frac{d\alpha }{dt}=k(1-\alpha )$$

The reaction rate constant *k* is represented by the Arrhenius equation as [44]:(2)k=Aexp(−E/RT)
where *A* (min^−1^) stands for the pre-exponential factor, and *E* (kJ·mol^−1^) represents the reaction activation energy. *R* is the gas constant, 8.314 J/(mol·K), and *T*(K) is the thermal temperature.

Besides, α is the reaction conversion rate and is calculated by the following equation:(3)$$\alpha =\frac{m_{0}-m_{t}}{m_{0}-m_{f}}\times 100\%$$
where *m*_0_ is the initial mass (g) of the sample, m_t_ is the sample mass (g) at the time point of the reaction t, and *m*_f_ is the final mass (g) following the reaction.

To substitute the heating rate constant β=dT/dt for the formula:(4)$$\frac{d\alpha }{dT}=\frac{A}{\beta }\exp(\frac{-E}{RT})\times \left(1-\alpha \right)$$

The coats-Redfern method was used to obtain [45]:

When the series is 1,
(5)$$\ln[\frac{-\ln(1-\alpha )}{T^{2}}]=\ln[\frac{AR}{\beta E}(1-\frac{2RT}{E})]-\frac{E}{RT}$$

For the vast majority of the pyrolysis reactions, $\frac{2RT}{E}\ll 1$, Equation (4) can be approximated by:(6)$$\ln[\frac{-\ln(1-\alpha )}{T^{2}}]=\ln\frac{AR}{\beta E}-\frac{E}{RT}$$

Because $\ln(\frac{AR}{\beta E})$ is constant in Equation (5), a straight line is drawn in $\ln[\frac{-\ln(1-\alpha )}{T^{2}}]$ against $\frac{1}{T}$. The reaction activation energy E and the pre-exponential factor A are available by drawing the slope $-\frac{E}{R}$ and intercept $\ln(\frac{AR}{\beta E})$ of the straight line.

### 2.4. Plotting and Statistical Analysis

Plotting and data analysis used Origin 2021 (OriginLab, Northampton, MA, USA) and SPSS19.0 (IBM Corporation, Armonk, NY, USA). The data were checked for normality and homogeneity of variance prior to doing the statistical analysis. If the variables were not normally distributed, nonparametric tests were used for the analysis. All experiments were conducted in triplicate for further statistical analysis.

## 3. Results and Discussion

### 3.1. Characterization of MS and PP

Table 1 shows the elemental analysis results of MS and PP. The results showed that both MS and PP had lower contents of S and N, which was beneficial to their thermochemical applications. In addition, the elemental compositions of MS and PP showed no significant differences in N and S. In addition, the lower S and N content prevented the production of toxic gases (SO_2_ and NO_x_). The high oxygen content of MS was demonstrated by the substantial proportion of C and O elements, the existence of cellulose, hemicellulose, and lignin, as well as oxygen-containing compounds. Hence, we assume the production of oxygenated organics. The C and H elements in PP accounted for a large proportion, which may lead to high heat value (HHV) [46]. Plastic (PP) has a relatively higher carbon content and predicted HHV compared to biomass (MS). The co-pyrolysis of PP and MS can effectively increase the content of hydrogen supplied by the pyrolysis reaction system and improve the resource utilization of MS. The elemental analysis of MS and PP was consistent with the literature [47,48]. Therefore, the co-pyrolysis process of MS and PP was feasible and efficient, and further studies were necessary.

### 3.2. Separate Pyrolysis of MS and PP

Figure 1a shows the TG-DTG curve of MS at the heating rate of 10 °C·min^−1^. The pyrolysis process of MS can be divided into three stages: the first phase was within room temperature to 200 °C, and the temperature range of 120 °C to 150 °C was the drying phase, and the MS was heated as dehydration occurred, while the chemical components did not change significantly. The weight loss rate was up to 4.349% due to water and the removal of low volatile compounds. The second stage occurred at 200–500 °C. The TG curve dropped sharply, which was the main stage of MS pyrolysis, with a weight loss rate of about 60.738. The DTG curve showed the largest weight rate peak, with the highest loss rate of −0.65%·min^−1^, corresponding to a temperature of 361.85 °C. The main reason was that the cellulose, hemicellulose, and lignin contained in MS were thermally decomposed at this stage, generating a large number of gas and liquid products, including CO, CO_2_, CH_4_, acetic acid, and methanol [49,50]. Actually, CO could be formed via the rupture of ether bridges [51]. The formation of CO_2_ was mainly due to the formation of organic carboxylic acids and alkyl groups from the cleavage of aliphatic hydrocarbon chains by equations as follows [52]. In the third stage, the mass loss of 22.849% was caused by lignin pyrolysis and generated more carbon, C-H, and C-C bonds break to form small molecules [53].
(7)$$R\text{-}\mathrm{COOH}↔R\text{-}H+CO_{2}$$
(8)$$R-{\mathrm{CH}}_{2}\text{-}{\mathrm{CH}}_{3}↔R-H+C_{3}H_{6}{\;\mathrm{or}\;R\text{-}\mathrm{CH}}_{3}{+C}_{2}H_{4}$$

The PP pyrolysis was different from the MS because there was only one peak on its DTG curve. The lowest fixed carbon content in PP, as indicated in Table 1, corresponding with the lowest residue of PP. Moreover, the pyrolysis process of PP had a higher temperature and a narrower range, mainly because the PP structure was simple and did not contain water. Figure 1b shows the TG-DTG curves of PP at a warming rate of 10 °C·min^−1^. At the first stage of PP pyrolysis (40–260 °C), the TG and DTG curves in this interval a straight line because PP was simply heated without obvious pyrolysis weight loss reaction, but meanwhile, the high polyolefin structure in PP would undergo dehydration, free radical fracture and other reactions. The second phase was 260–431 °C, in which PP started to depolymerize and corresponded to a dechlorination reaction, followed by conjugated double bonds, with the formation of a small number of hydrocarbons [54]. This phase was the primary reaction phase of PP pyrolysis, and it had a maximum weightlessness peak that occurred around 379 °C, with a maximum weight loss rate of 1.154%·min^−1^. After the second stage, the pyrolysis residues of PP samples were nearly zero, indicating that PP was the main free radical pyrolysis, and the resulting hydrocarbons were mainly broken from the C-C bond rather than the C-H bond. The pyrolysis process of PP can be simply summarized as one-step pyrolysis. From the beginning to the end of the reaction, the process of dissolving the polymer structure was simply used to generate the volatile products [48,55,56]. The weight loss rate of the whole pyrolysis process was 99.3%.

From the point of view of thermogravimetric characteristics, the DTG peak of MS appears earlier than that of PP, which indicates that more energy input is required during pyrolysis. In addition, Figure 1 shows that the pyrolysis process of PP is simpler, and the DTG peak is narrower compared with MS, indicating that the reaction process may be easier to proceed than that of MS.

### 3.3. Co-Pyrolysis of SP11

Figure 2 presents the TG-DTG curves for the pyrolysis of the mixtures of MS and PP in a weight ratio of 1:1 at a heating rate of 10 °C·min^−1^. MS co-pyrolysis with PP can be divided into three stages.

The first stage was from room temperature to 200° C, and the reaction occurred when the MS lost water and showed a small weight loss peak on the DTG curve. The second stage was the main phase of the weight loss of the mixed sample, with a maximum weight loss rate of 82%, ranging from 200 °C to 510 °C. According to the DTG curve, two weight loss peaks appeared at this stage, and the first weight loss peak appeared between 200 °C and 394 °C, mainly due to the weight loss of MS. The temperature corresponding to the peak value was 367 °C, and the weight loss rate was 3.53%·min^−1^. The second weight loss peak appeared between 394 °C and 510 °C. Since the pyrolysis temperature of MS coincided with that of PP, this stage included the weight loss process of MS and PP. The peak temperature was 485 °C, and the weight loss rate was 8.25%·min^−1^. A similar phenomenon was also observed in the co-pyrolysis of municipal paper and polypropylene waste [57].

The obvious interaction between MS and PP after mixing and co-pyrolysis expanded the reaction temperature interval of pyrolysis, and co-pyrolysis enhanced the temperature difference between the two maximum weightlessness peaks corresponding to MS and PP pyrolysis, respectively. The peak of the corresponding plastic component pyrolysis in SP11 was significantly shifted towards the high-temperature side. The possible reason was that the biomass component (such as MS) pyrolysis in SP11 occurred earlier than the corresponding component decomposition in PP pyrolysis, resulting in the temperature increase corresponding to the peak of plastic component pyrolysis [58].

The cumulative amount of CO_2_ derived from biochars was significantly decreased with pyrolysis temperature (*p* < 0.05), indicating that biochars prepared at higher temperatures were more stable in the soil [59] as can be seen from Figure 2, with the increase in heating rate, the TG curve moves to the high temperature region. This indicates that MS and PP are more conducive to the generation of biochar in the process of rapid pyrolysis and high temperature. Moreover, Chen et al. (2016) found that during the co-pyrolysis of wood and PVC, the yield of biochar was much higher than the predicted value. In addition, the co-pyrolysis of wood and PP also improved the inhomogeneity of biochar [60]. In addition, high pyrolysis temperatures can also increase the surface area and carbonized percentage of biochar, which improves the substance’s ability to bind both inorganic and organic contaminants to contaminated soil [61]. In addition, biochar, a solid by-product of biomass pyrolysis, used as a soil supplement in agriculture has been suggested. Biochar has been shown to induce changes in soil microbial communities. The addition of biochar to some plant growth media can cause chemical changes that affect the rhizosphere microbiome and trigger a variety of responses in the plant, as well as can be beneficial for plant growth [62]. Therefore, the technological intervention is well received by the community.

Temperature is a crucial element affecting the liberation of sulfur and nitrogen. At 300–600 °C, COS and SO_2_ are often released, but HCN and NH_3_ are frequently released at 700–900 °C. The amount of sulfur and nitrogen still present in the solid diminishes as the temperature rises. For conventional gases, the release order of gases during MS and PP blend pyrolysis is H_2_O, CO_2_, CO, CH_4_, and H_2_. Thus, in applications where biochar with low sulfur and low nitrogen content is required, pyrolysis should occur at higher temperatures [63]. In addition, co-pyrolysis can greatly improve the yield of bio-oil, promote the formation of long-chain fatty acids, and inhibit the formation of acetic acid, O-containing substances, phenols, and N-containing compounds, the quality of bio-oil was significantly improved. In addition, co-pyrolysis increases the yield of N_char_ and N_gas_ and decreases the yield of N_bio-oil_, thus promoting nitrogen conversion [64].

### 3.4. Thermogravimetric Characteristics and Kinetic Analysis

#### 3.4.1. TG-DTG Curves at Different Heating Rates

The heating rate was an important factor affecting the pyrolysis of MS and PP [65]. Figure 3 shows the TG-DTG curve of MS pyrolysis at different heating rates. As the heating rate increased, the TG curve moved towards the high-temperature region. At the same temperature, the faster the heating rate, the lower the heat loss rate. The DTG curve presented that the heat loss rate increases with the faster heating rate and the maximum weight loss peak of the DTG curve gradually moved to the high-temperature region. The data indicated that accelerating the heat loss rate contributed to the rapid progress of pyrolysis. The thermal weight loss rate of MS was basically the same at the heating rate of 10 °C·min^−1^ and 20 °C·min^−1^, but the weight loss rate of the heating rate of 30 °C·min^−1^ was less than that of the other two groups, which was due to the increase in heating rate leads to the intensification of thermal hysteresis, which led to the incomplete reaction of the whole pyrolysis process [66].

Figure 4 shows the TG-DTG curves for PP pyrolysis at different heating rates. As with the MS curve trend, the overall TG curve moved towards the high temperature as the heating rate increased. However, at the beginning and the end, with the increase in temperature, the weight loss rate of the final sample was not much different at the same temperature as the PP pyrolysis interval, the faster the heating rate, the less thermal weight loss rate. The characteristics of the DTG curves were also similar to the MS.

Nevertheless, the co-pyrolysis of the MS and PP mixture was quite different from the pyrolysis alone. Figure 5 shows the TG-DTG curves for the co-pyrolysis of MS and PP at different heating rates. When the heating rate increased, the TG curve moved slightly to the high-temperature region, and the sample weight loss rate decreased. This phenomenon is observed for many raw materials [67]. With the acceleration of the pyrolysis rate, the pyrolysis reaction time was shortened, and the sample could not react completely, so there were more solid residues. The mass losses of mixtures at β = 10, 20, and 40 °C/min are 94.34%, 94.33%, and 93.52%, respectively. The mass loss did not change significantly with the increase in the heating rate. This means that the increase in the heating rate has less effect on the mass loss of the mixture. The changing trend of the DTG curve was that the heat loss rate increased as the heating rate accelerated, and the pyrolysis start temperature and germination temperature of the main pyrolysis range moved to the high-temperature zone [24,68,69]. In fact, increasing the heating rate tended to delay the pyrolysis process of the mixture and the raw material will decompose at higher temperatures [70].

In order to explore whether pyrolysis cooperated between MS and discarded meal boxes at different heating rates, the theoretical weight mass of the heating rate at 10 °C·min^−1^, 20 °C·min^−1^, and 30 °C·min^−1^ was calculated by Equation (6) and compared with the experimental TG curve, as shown in Figure 6.
(9)$$W_{\mathrm{Cal}}{=x}_{\mathrm{MS}}W_{\mathrm{MS}}{+x}_{\mathrm{PP}}W_{\mathrm{PP}}$$
where, W_Cal_ is the theoretical weight loss of the sample, %; $x_{\mathrm{MS}}$ and $x_{\mathrm{PP}}$ are the ratio of MS and PP in the mixed sample,%; W_MS_ and W_PP_ are the weight loss of MS and PP at corresponding temperatures, %.

According to Figure 6, the experimental weight loss curve and theoretical weight loss curve of SP11 did not coincide, indicating that there was a synergistic effect in the co-pyrolysis process of MS and discarded meal boxes and affected the pyrolysis characteristics of the sample. When the heating rate was 10 °C·min^−1^ and 20 °C·min^−1^, the theoretical and actual TG curves overlapped roughly, revealing that there was no significant synergistic effect under this condition. When the heating rate was 30 °C·min^−1^, the experimental curve of TG showed a larger weight loss and less solid residue than the theoretical curve, demonstrating that co-pyrolysis produced some degree of synergistic effect at this warming rate. The presence of PP in the co-pyrolysis process promoted the MS pyrolysis because the solid residue presented after PP alone pyrolysis was very small, only 0.7%.

Biomass co-pyrolysis with plastics appeared to have a synergistic effect; however, it has not been properly investigated. Studies generally agree that the pyrolysis of polyolefin and biomass is a free radical reaction. When biomass and plastics are co-pyrolyzed, the biomass is first decomposed to produce free radicals, which triggers the polyolefin chain decomposition reaction. Plastic contains about 16% hydrogen. During the co-pyrolysis of plastic and biomass, hydrogen can be supplied to biomass, then stabilizing free radicals generated by biomass decomposition, forming volatile substances, increasing weight loss rate, and reducing char [71].

#### 3.4.2. Kinetic Analysis

A kinetic analysis of MS, PP, and SP11 was performed based on the first-level model, focusing on the main decomposition phase. The corresponding parameters are summarized in Table 2. The results showed that the correlation index (*R*^2^) of the first-order reaction models of all samples was greater than 0.92, which reveals that the reaction model adopted can describe all pyrolysis processes well.

Changes in kinetic parameters may indicate a shift in the mechanism governing the pyrolysis process, according to the fundamental principles of dynamics. Table 2 listed that the kinetic parameters might significantly differ depending on the varied heating rates. Additionally, the activation energy change law for various heating rates is shown in Table 2. The data in Table 2 demonstrated that the activation energy of PP decomposition was higher than that of biomass decomposition. This was due to the large difference in the molecular structure between biomass and plastic and the much higher energy barrier of plastic [72]. In the higher temperature range, as well as at the rate of temperature increase, the activation energy was also higher, meaning that the pyrolysis reaction was more difficult to carry out. The activation energy of MS ranged from 42.7 to 47.4 KJ·mol^−1^ and was dependent on the heating rate, which was significantly lower than that of woody biomass, such as sesame straw and almond shells, indicating that MS biomass was more easily destroyed under warmer temperatures [73]. The E of PP (107.7 kJ·mol^−1^) is higher than that of MS (42.7 kJ·mol^−1^), corresponding to the experiments that the peak pyrolysis temperature of PP is approximately 70 °C which is higher than the MS. It has been proposed that at lower temperatures, the free radicals produced by biomass degradation promote polymer decomposition and hence lower activation energies are obtained [74]. Moreover, from Table 2, the activation energy of the mixture of MS and PP in the first stage is reduced by about 60% compared with the pyrolysis of PP individuals. Burra et al. (2018) determined the apparent activation energy of PP was 144.55 KJ·mol^−1^, which was nearly identical to the result we measured. MS and PP co-pyrolysis can lower the PP activation energy, making the pyrolysis easier to carry out [75]. Table 2 also demonstrated that MS and PP co-pyrolysis and MS and PP pyrolysis alone were comparable. As the heating rate increased, the activation energy of PP increased, resulting in a more difficult pyrolysis reaction. Higher apparent activation energy meant that higher energy was needed to break the bonds, which made the reaction slower [76].

In this paper, the activation energy obtained by co-pyrolysis of MS and PP was 43.3 KJ·mol^−1^ at a heating rate of 10 °C·min^−1^ with a temperature range of 230–520 °C. Table 3 lists the thermo kinetic parameters obtained after co-pyrolysis of other samples mixed with PP from the other works of literature. The mixture temperature range and activation energy of pyrolysis were no longer competitive compared with SP11, and the activation energies of these mixtures were much larger than those of SP11. Therefore, MS and PP co-pyrolysis improved the total pyrolysis temperature range and lowered the activation energy of the reaction more thoroughly. Meanwhile, this study helped to understand the role of MS and plastic on the co-pyrolysis process, including improving the product quality and reducing the activation energy and solid residues, which could provide a practical and efficient method for future co-pyrolysis application, and was attractive for engineering applications.

## 4. Conclusions

The TG-DTG analysis of MS, PP, and SP11 was explored in the current study. SP11 presented obvious interaction in the pyrolysis process. The presence of plastics could contribute to the degradation of MS biomass resources. The higher hydrogen content of the plastic can balance the oxygen content of the product when co-pyrolyzing with MS, improving the pyrolysis effect. In addition, co-pyrolysis expanded the reaction temperature range of pyrolysis, and the peak value moved to the high-temperature side. When the heating rate was 30 °C·min^−1^, the weight loss rate of the TG experimental curve was larger than that of the TG theoretical curve, resulting in the reduction in solid residues, which indicated that co-pyrolysis produced a certain degree of synergistic effect at this heating rate. At the same time, through kinetic analysis, the co-pyrolysis of MS and PP reduced the activation energy, and the lowest value of activation energy was 43.3 J·mol^−1^. Compared with MS, the SP11 activation energy was not different, but compared with PP, the SP11 activation energy was reduced by about 60%. The results illustrated that MS was a promising biomass for co-pyrolysis with plastic, and co-pyrolysis led to the reaction being easier and improved the pyrolysis efficiency.

## Figures and Tables

**Figure 1 foods-11-03840-f001:**
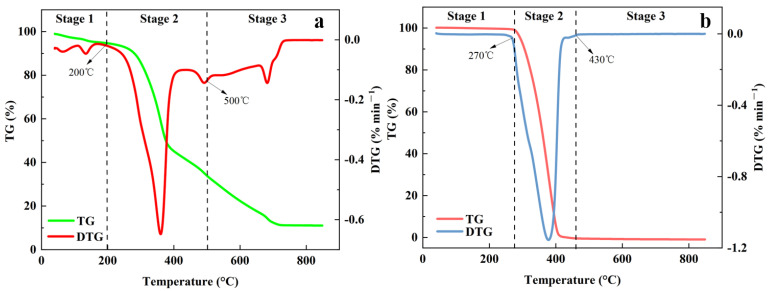
(**a**) TG-DTG curve of pyrolysis of MS at a heating rate of 10 °C·min^−1^; (**b**) TG-DTG curve of pyrolysis of PP at a heating rate of 10 °C·min^−1^. TG: thermal weight curve, DTG: differential thermal weight curve.

**Figure 2 foods-11-03840-f002:**
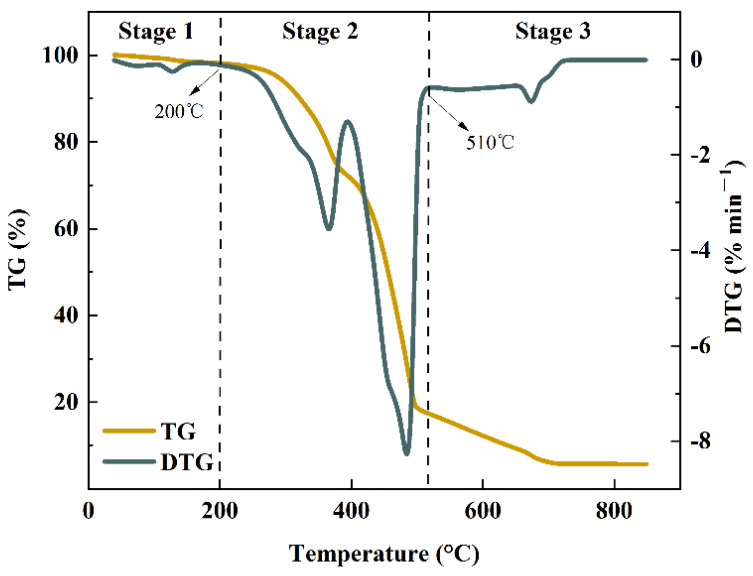
TG-DTG curve of pyrolysis of MS-PP mixtures at a heating rate of 10 °C·min^−1^.

**Figure 3 foods-11-03840-f003:**
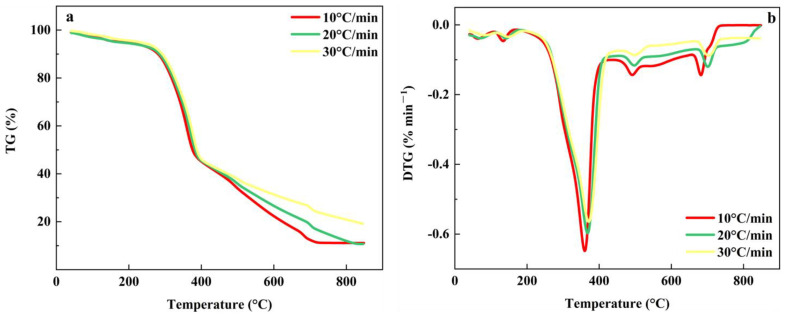
The TG-DTG curve for MS pyrolysis at different heating rates. (**a**) The TG curve of MS; (**b**) The DTG curve of MS.

**Figure 4 foods-11-03840-f004:**
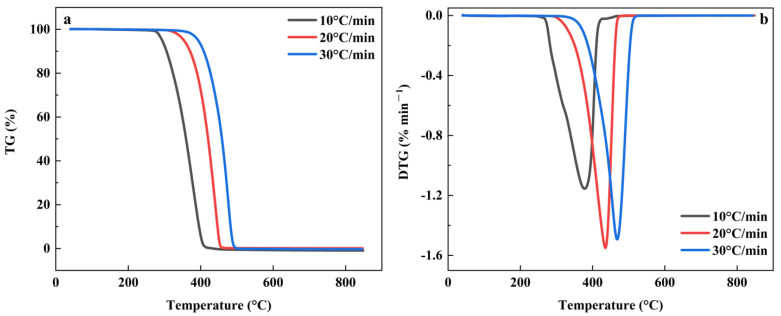
The TG-DTG curves for PP pyrolysis at different heating rates. (**a**) The TG curve of PP; (**b**) The DTG curve of PP.

**Figure 5 foods-11-03840-f005:**
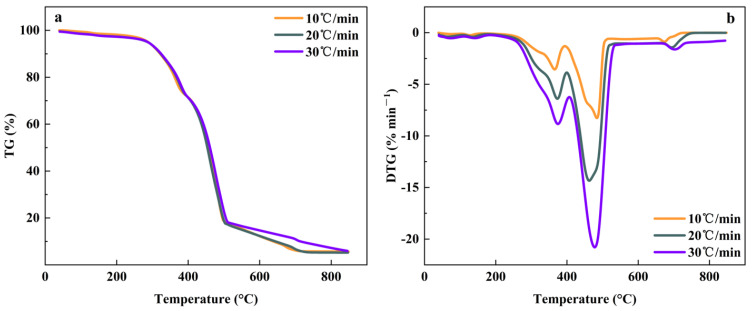
(**a**) The TG curves of co-pyrolysis of MS and PP at different heating rates; (**b**) The DTG curves of co-pyrolysis of MS and PP at different heating rates.

**Figure 6 foods-11-03840-f006:**
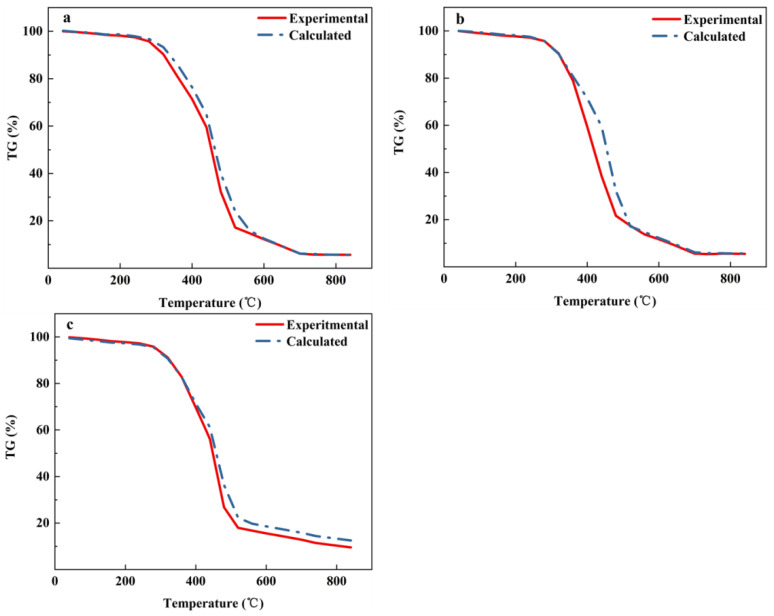
Theoretical and practical TG curves for the co-pyrolysis of MS and PP at different heating rates. (**a**) 10 °C·min^−1^; (**b**) 20 °C·min^−1^; (**c**) 30 °C·min^−1^.

**Table 1 foods-11-03840-t001:** Elemental analysis of MS and PP.

Sample	Elemental Analysis (%)				
	C	H	O	N	S
MS	37.73	5.38	53.59	2.09	0.18
PP	81.65	16.71	0.94	0.00	0.37

MS: Mushroom sticks; PP: Polypropylene.

**Table 2 foods-11-03840-t002:** Pyrolysis kinetic parameters of MS, PP, and SP11 at different heating rates.

Sample	Heating Rate (°C·min^−1^)	Temperature Range (°C)	Activation Energy (E) (kJ·mol^−1^)	Frequency Factor (A) (mol^−1^)	Correlation Index (R^2^)
	10	200–500	42.7	246.65	0.971
MS	20	245–401	43	472.12	0.978
	30	249–408	47.4	1901.80	0.986
	10	260–431	107.7	1.8 × 10^8^	0.991
PP	20	300–465	141.4	2.6 × 10^10^	0.996
	30	340–490	150.3	4.1 × 10^10^	0.996
SP11	10	230–520	43.3	103.58	0.980
	20	261–521	45.1	303.37	0.982
	30	270–520	44.41	377.91	0.983

**Table 3 foods-11-03840-t003:** Pyrolysis kinetic parameters of other samples co-pyrolysis with PP.

Sample	Temperature Range (°C)	Activation Energy (E) (kJ·mol^−1^)	Frequency Factor (A) (mol^−1^)	Correlation Index (R)	Reference
	250–348	60.44	1.43 × 10^4^	0.988	[48]
SP-PP	440–520	129.29	5.68 × 10^8^	0.919	
	-	-	-	-	
PP-D. tertiolecta (6:4)	-	-	-	-	[77]
	-	142.9	2.68 × 10^13^	-	
	-	-	-	-	
PP-Tyre	319–378	70.08	1.85 × 10^4^	0.998	[78]
	378–428	52.08	5.19 × 10^2^	0.997	
	428–488	138.04	2.94 × 10^9^	0.971	
DS-PP	224–295	49.8	2.6 × 10^2^	0.980	[27]
	295–345	45.7	1.0 × 10^3^	0.980	
	446–504	160.1	1.8 × 10^11^	0.960	
SW-PP (3:7)	286–405	128	1.77 × 10^10^	0.978	[72]
	405–461	249	1.26 × 10^18^	0.976	
	461–525	426	1.19 × 10^29^	0.982	
95%LVC-PP (1:5)	191–399	35.7	351	0.945	[79]
	399–474	242.4	3.0 × 10^17^	0.988	
	474–516	558.8	7.7 × 10^32^	0.999	
	516–617	198.4	9.5 × 10^11^	0.968	
Bamboo-PP (4:1)	-	-	-	-	[80]
	-	213.81	2.87 × 10^17^	-	
	-	-	-	-	

‘-’ represents ‘not reported’; ‘DS’ represents ‘Dunaliella salina’; ‘SW’ represents ‘solid waste’; ‘LVC’ represents ‘low volatile coal’.

## Data Availability

Data is contained within the article.

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
