# Peer review of "Sustainable Environmental Assessment of Waste-to-Energy Practices: Co-Pyrolysis of Food Waste and Discarded Meal Boxes"

_foods, 2022, doi:10.3390/foods11233840_

Round 1

Reviewer 1 Report

Hi dear

This article "Thermogravimetric characteristics and kinetics of the co-pyrolysis of mushroom sticks and discarded meal boxes” was revised and has a novelty and I recommend it for consideration of the following comments.

Title: It is perfect.

Abstract:

·       The type of statistical design used in this research should be mentioned.

·       It is really perfect.

Keywords: It is perfect and good selected.

Abbreviation:

·       Please provide “Abbreviation section consequent the Keywords

Introduction:

·       Please provide the references according to guideline for the authors either in the manuscript or in the reference list.

·       Use food waste as an enrichment in food formulation as an example in the introduction.

Materials:

·                 Please write materials as Company Name (City, Country), especially for chemical analysis assessment which used in the study.

·       MS was from Pingquan City, Hebei Province. It is not good expression. Please amended it.

Methodology:

·               Statistical analysis must be mentioned.

·               All the tables and figures must be self-explanatory for the better conception of the article reader.

 Results:

·       Table 2, and 3: Please consider the statistical analysis comparing so that the alphabetical statistical letters for the means should all be modified such that the greatest number has the letter a and as the numbers go lower, letters b, c etc.

Discussion:

Discussion text must grammar improve and in some cases it is very weak and maybe there is no discussion at all.

Conclusions:

Conclusion is perfect.

References: It is OK.

The article has a little flaws in express and concept of English, it is suggested to be revised in a scientific and native way.

Author Response

Responses to Reviewers

Manuscript Number: foods-1886527

Answers to Editor and Reviewers.

The revised manuscript was earlier submitted for publication in Foods. We are obliged to the reviewers of this manuscript, which contributed to improve its quality. We carefully revised the manuscript according to the comments. Relevant changes (track changes) were introduced throughout the manuscript text and marked with line number in this file.

This article "Thermogravimetric characteristics and kinetics of the co-pyrolysis of mushroom sticks and discarded meal boxes” was revised and has a novelty and I recommend it for consideration of the following comments.

1) The type of statistical design used in this research should be mentioned.

Response: Thanks for the critical comments. The type of statistical design was added as the abstract. The relevant text was revised to ‘Experimental study was employed using response surface methodology-Box-Behnken (RSM-BB). The statistical analysis showed that the factors of pyrolysis temperature and particle size had the greater effect, while the heating rate was significant’.(Lines 16-19)

2) Please provide “Abbreviation section consequent the Keywords.

Response: We appreciate your suggestion very much. Accordingly, we have added ‘Abbreviation section’ consequent the ‘Keywords’ (see below).

Abbreviation

MS

mushroom sticks

t

time

PP

polypropylene

E

activation energy

SP11

mixture of MS and PP

f(α)

reaction mechanism function

TG-FTIR

thermogravimetric analyzer coupled with Fourier transform infrared spectroscopy

A

pre-exponential factor

WCal

theoretical weight loss of the sample

T

reaction temperature

xMS

ratio of MS in the mixed sample

R

universal gas constant (8.314 J/(mol·K))

WPP

weight loss of PP at corresponding temperatures

m0

the initial mass (g) of the sample

WMS

weight loss of MS at corresponding temperatures

mt

the sample mass (g) at the time point of the reaction t

α

mass conversion rate

TG

thermal weight curve

DS

Dunaliella salina

DTG

differential thermal weight curve

SW

solid waste

LVC

low volatile coal

3) Please provide the references according to guideline for the authors either in the manuscript or in the reference list. Use food waste as enrichment in food formulation as an example in the introduction.

Response: We appreciate this kind recommendation. The relevant text was revised to ‘In addition, the value of food waste (such as fruit and vegetable waste) provides an opportunity to use the extraction of food waste and to develop valuable functional foods[6]’ in the introduction.(Lines 56-58)

4) Please write materials as Company Name (City, Country), especially for chemical analysis assessment which used in the study. MS was from Pingquan City, Hebei Province. It is not good expression. Please amended it.

Response: Thanks for the comments. We had amended source of materials in Methods and Materials. The relevant text was revised to ‘MS was obtained from Huixin Mushroom Industry Co., LTD (Pingquan, China)’.(Lines 125-126)

5) Statistical analysis must be mentioned. All the tables and figures must be self-explanatory for the better conception of the article reader.

Response: Thanks for the critical comments. The relevant text was added to ‘2.4 Plotting and Statistical Analysis. Data were analysed and plotted using the Origin 2021 (OriginLab, Northampton, USA) and SPSS 19.0 (IBM Corporation, Armonk, USA). The data normality and homogeneity of variance were checked prior to the statistical analysis. If the variables were not normally distributed, nonparametric tests were used for the analysis. All experiments were conducted in triplicate’ in the section of Materials and Methods.(Lines 162-167)

All the tables and figures had modified self-explanatory for the better conception of the article reader. The relevant text was revised to ‘Figure 1. a) TG-DTG curve of pyrolysis of MS at a heating rate of 10°C•min-1; b) TG-DTG curve of pyrolysis of PP at a heating rate of 10°C•min-1. Figure 5. a) The TG-DTG curves of co-pyrolysis of MS and PP at different heating rates; . The TG-DTG curves of co-pyrolysis of PP at different heating rates. Table 3. Note: ‘-‘represents ‘not reported’; ‘DS’ represents ‘Dunaliella salina’; ‘SW’ represents ‘solid waste’; ‘LVC’ represents ‘low volatile coal’.(Lines 234-235, 309-310, 375-376)

6) Table 2, and 3: Please consider the statistical analysis comparing so that the alphabetical statistical letters for the means should all be modified such that the greatest number has the letter a and as the numbers go lower, letters b, c

Response: Thanks for the critical comments, but we don’t think the Table 2 and Table 3 are not suitable for statistical analysis. The reasons are as follows: the data in Table 2 and Table 3 are fixed values calculated according to the first-order kinetic model, and this research mainly focuses on the thermogravimetric analysis of mushroom sticks and waste plastic polypropylene. Pyrolysis product analysis is not quite meaningful to this paper mainly because this paper focuses on the co-pyrolysis of MS and PP. The TG-DTG obtained by TG-DTG and the first-order reaction kinetic model are used to solve the activation energy. The obtained results provide a theoretical basis for the recycling and energy utilization of MS and discarded meal boxes.

7) Discussion text must grammar improve and in some cases it is very weak and maybe there is no discussion at all.

Response: Thanks for the suggestion. More relevant information from the literature was collected to support the current finding. Moreover, more following discussion was added in the text.

3.1. Characterization of MS and PP.

And the elemental compositions of MS and PP showed no significant differences in N and S.(Lines 171-172)

Hence we assume the production of oxygenated organics.(Lines 176-177)

The C and H elements in PP accounted for a large proportion, which may led to high heat value (HHV) [40]. Plastic (PP) had a relatively higher carbon content and predicted HHV compared to biomass (MS). The co-pyrolysis of PP and MS can effectively increase the content of hydrogen supplied by the pyrolysis reaction system and improve the resource utilization of MS. The elemental analysis of MS and PP were consistent with the literature [41,42]. Therefore, the co-pyrolysis process of MS and PP was feasible and efficient, and further studies were necessary.(Lines 177-184)

3.2. Separate Pyrolysis of MS and PP.

Actually, CO could be formed via rupture of ether bridges [45]. The formation of CO2 was mainly due to the formation of organic carboxylic acids and alkyl groups from the cleavage of aliphatic hydrocarbon chains by equations as follows [46].(Lines 199-202)

 The lowest fixed carbon content in PP, as indicated in Table 1, corresponded with the lowest residue of PP. (Lines 208-209). From the point of view of thermogravimetric characteristics, the DTG peak of MS appeared earlier than that of PP, which indicated that more energy input was required during pyrolysis. In addition, Fig.1 shows that the pyrolysis process of PP is simpler and the DTG peak is narrower compared with MS, indicating that the reaction process may be easier to proceed than that of MS.

3.3. Co-pyrolysis of SP11.

A similar phenomenon was also observed in the co-pyrolysis of municipal paper and polypropylene waste [52].(Lines 249-251)

3.4.1. TG-DTG curves at different heating rates.

This phenomenon is observed from many raw materials [56].(Lines 290-291). The mass losses of mixtures at β = 10, 20 and 40 °C /min are 94.34%, 94.33% and 93.52%, respectively. The mass loss did not change significantly with the increase of heating rate. This means that the increase in the heating rate has less effect on the mass loss of the mixture (Lines 293-296) In fact, increasing the heating rate tended to delay the pyrolysis process of the mixture and the raw material will decompose at higher temperatures.(Lines 299-301)

3.4.2. Kinetic Analysis.

The results showed that the correlation indices (R2) of the first-order reaction models of all samples are greater than 0.92, which indicates that the reaction model adopted can describe all pyrolysis processes well.(Lines 343-345). The data in Table 2 demonstrated that the activation energy of PP decomposition was higher than that of biomass decomposition. This was due to the large difference in the molecular structure between biomass and plastic, and a much higher energy barrier of plastic [60].(Lines 351-354)The E of PP (107.7 kJ•mol-1) is higher than that of MS (42.7 kJ•mol-1), corresponding to the experiments that the peak pyrolysis temperature of PP is approximately 70 °C which is higher than the MS. It has been proposed that at lower temperatures, the free radicals produced by biomass degradation promote polymer decomposition, hence lower activation energies are obtained [62]. Moreover, from Table 2, the activation energy of the mixture of MS and PP in the first stage is reduced by about 60% compared with the pyrolysis of PP individual. (Lines 359-366)

Meanwhile, this study helped to understand the role of MS and plastic on the co-pyrolysis process, including improving the product quality and reducing the activation energy and solid residues, which could provide a practical and efficient method for future co-pyrolysis application, and was attractive for engineering applications. (Lines 384-388)

8) The article has a little flaws in express and concept of English, it is suggested to be revised in a scientific and native way.

Response: We apologize for the unsatisfying language and have worked on the manuscript language and readability. The manuscript was modified by the native English speaker for language corrections. We really hope that the writing has been substantially improved.

Reviewer 2 Report

Dear authors,

After checking carefully your MS, although I consider it is really interesting and the topic time relevant, unfortunately, taking into account the aims and scope of Foods I consider it is not the best choice for this kind of paper.

It is true it is covering mushroom sticks and discarded meal boxes, which can be considered as a food waste but the topic could fit better with a related journal.

Author Response

The revised manuscript was earlier submitted for publication in Foods. We are obliged to the reviewers of this manuscript, which contributed to improve its quality. We carefully revised the manuscript according to the comments. Relevant changes (track changes) were introduced throughout the manuscript text and marked with line number in this file.

After checking carefully your MS, although I consider it is really interesting and the topic time relevant, unfortunately, taking into account the aims and scope of Foods. I consider it is not the best choice for this kind of paper. It is true it is covering mushroom sticks and discarded meal boxes, which can be considered as a food waste but the topic could fit better with a related journal.

Response: Thanks for the critical comments. We understand why reviewer doesn’t think this manuscript is quite suitable for the aims and scopes of Foods. Although the article is not a typical food-focused article but in a broader scope, the article addresses very important aspects of our food system for a sustainable future.  Minimizing food waste or developing strategies to utilize food waste and packaging materials or meal boxes are integral to our food system. Therefore, I believe this manuscript should be received reconsideration in Foods.

Reviewer 3 Report

The manuscript entitles “Thermogravimetric characteristics and kinetics of the co-pyrolysis of mushroom sticks and discarded meal boxes” is an interesting manuscript that aims to explore characterization of underutilized mushroom sticks and meal boxes. It is good to see a selection of studies based on practical issue. The section wise comments are below;
Abstract. In abstract background, findings and conclusion of study is presented nicely  
Introduction. The main idea/concept is explained in the introduced section and fully described with the background with relevant literature support.
Materials and methods. In this section all the procedures are written with required details except only 1st paragraph need improvement as grammatically it is not correct.
3. The results are explained nicely with Tables, graphs and equations to understand the in depth knowledge. Discussion is supported with the relevant literature.
Moreover, the conclusion is convincing enough for the study and its findings.

Reviewer 4 Report

Here I would like to send my comments about the manuscript "Thermogravimetric characteristics and kinetics of the co-pyrolysis of mushroom sticks and discarded meal boxes". The paper submitted in FOODS was written with good quality. The methodology is objective and very well described. The results are interesting, and the discussion well-structured. The conclusion describes the work developed. The figures and tables are consistent with the work. I am favourable to minor revision.

1. The author should include a clear hypothesis in the introduction.

2. The author could post a photo of biomass and plastic before and after pyrolysis and SEM photo.

3. Why choose microorganisms Mushroom sticks? What are the advantages compared to other microorganism? Please point this out in the text.
